# Learning Transferrable Representations for Unsupervised Domain Adaptation

**Ozan Sener**[1], **Hyun Oh Song**[1], **Ashutosh Saxena**[2], **Silvio Savarese**[1]

Stanford University[1]    Brain of Things[2]

{ozan,hsong,asaxena,ssilvio}@cs.stanford.edu

## Abstract

Supervised learning with large scale labelled datasets and deep layered models has caused a paradigm shift in diverse areas in learning and recognition. However, this approach still suffers from generalization issues under the presence of a domain shift between the training and the test data distribution. Since unsupervised domain adaptation algorithms directly address this domain shift problem between a labelled source dataset and an unlabelled target dataset, recent papers [11, 33] have shown promising results by fine-tuning the networks with domain adaptation loss functions which try to align the mismatch between the training and testing data distributions.

Nevertheless, these recent deep learning based domain adaptation approaches still suffer from issues such as high sensitivity to the gradient reversal hyperparameters [11] and overfitting during the fine-tuning stage. In this paper, we propose a unified deep learning framework where the representation, cross domain transformation, and target label inference are all jointly optimized in an end-to-end fashion for unsupervised domain adaptation. Our experiments show that the proposed method significantly outperforms state-of-the-art algorithms in both object recognition and digit classification experiments by a large margin.

## 1   Introduction

Recently, deep convolutional neural networks [17, 26, 30] have propelled unprecedented advances in artificial intelligence including object recognition, speech recognition, and image captioning. Although these networks are very good at learning state of the art feature representations and recognizing discriminative patterns, one major drawback is that the network requires huge amounts of labelled training data to fit millions of parameters in the complex network. However, creating such datasets with complete annotations is not only tedious and error prone, but also extremely costly. In this regard, the research community has proposed different mechanisms such as semi-supervised learning [27, 37], transfer learning [23, 31], weakly labelled learning, and domain adaptation. Among these approaches, domain adaptation is one of the most appealing techniques when a fully annotated dataset (e.g. ImageNet [7], Sports1M [14]) is already available as a reference.

The goal of unsupervised domain adaptation, in particular, is as follows. Given a fully labeled source dataset and an unlabeled target dataset, to learn a model which can generalize to the target domain while taking the domain shift across the datasets into account. The majority of the literature [13, 29, 9, 28, 32] in unsupervised domain adaptation formulates a learning problem where the task is to find a transformation matrix to align the labelled source data distribution to the unlabelled target data distribution. Although these approaches have shown promising results, they show accuracy degradation because of the discrepancy between the learning procedure and the actual target inference procedure. In this paper, we aim to address this issue by incorporating the unknown target labels into the training procedure.

In this regard, we formulate a unified deep learning framework where the feature representation, domain transformation, and target labels are all jointly optimized in an end-to-end fashion. The proposed framework first takes as input a batch of labelled source and unlabelled target examples, and maps this batch of raw input examples into a deep representation. Then, the framework computes the loss of the input batch based on a two stage optimization in which it alternates between inferring the labels of the target examples transductively and optimizing the domain transformation parameters.

Concretely, in the transduction stage, given the fixed domain transform parameter, we jointly infer all target labels by solving a discrete multi-label energy minimization problem. In the adaptation stage, given a fixed target label assignment, we seek to find the optimal asymmetric metric between the source and the target data. The advantage of our method is that we can jointly learn the optimal feature representation and the optimal domain transformation parameter, which are aware of the subsequent transductive inference procedure.

Following the standard evaluation protocol in the unsupervised domain adaptation community, we evaluate our method on the digit classification task using MNIST [19] and SVHN[21] as well as the object recognition task using the Office [25] dataset, and demonstrate state of the art performance in comparison to all existing unsupervised domain adaptation methods. Learned models and the source code can be reached from the project webpage `http://cvgl.stanford.edu/transductive_adaptation`.

## 2    Related Work

This paper is closely related to two active research areas: (1) Unsupervised domain adaptation, and (2) Transductive learning.

**Unsupervised domain adaptation**: [16] casts the zero-shot learning [22] problem as an unsupervised domain adaptation problem in the dictionary learning and sparse coding framework, assuming access to additional attribute information. Recently, [3] proposed the active nearest neighbor algorithm, which combines the component of active learning into the domain adaptation problem and makes a bounded number of active queries to users. Also, [13, 9, 28] proposed subspace alignment based approaches to unsupervised domain adaptation where the task is to learn a joint transformation and projection in which the difference between the source and the target covariance is minimized. However, these methods learn the transformation matrices on the whole source and target dataset without utilizing the source labels.

[32] utilizes a local max margin metric learning objective [35] to first assign the target labels with the nearest neighbor scheme and then learn a distance metric to enforce the negative pairwise distances to be larger than the positive pairwise distances. However, this method learns a symmetric distance matrix shared by both the source and the target domains so the method is susceptible to the discrepancies between the source and the target distributions. Recently, [11, 33] proposed a deep learning based method to learn domain invariant features by providing the reversed gradient signal from the binary domain classifiers. Although this method performs better than aforementioned approaches, their accuracy is limited since domain invariance does not necessarily imply discriminative features in the target domain.

**Transductive learning**: In the transductive learning [10], the model has access to unlabelled test samples during training. [24] utilizes a semi-supervised label propagation algorithm into the semi-supervised transfer learning problem assuming access to few labeled examples and additional human specified semantic knowledge. [15] tackled a classification problem where predictions are made jointly across all test examples in a transductive [10] setting. The method essentially enforces the notion that the true labels vary smoothly with respect to the input data. We extend this notion to jointly infer the labels of unsupervised target data points in a k-NN graph.

To summarize, our main contribution is to formulate an end-to-end deep learning framework where we learn the optimal feature representation, infer target labels via discrete energy minimization (*transduction*), and learn the transformation (*adaptation*) between source and target examples all jointly. Our experiments on digit classification using MNIST [19] and SVHN[21] as well as the object recognition experiments on Office [25] datasets show state of the art results, outperforming all existing methods by a substantial margin.

## 3 Method

### 3.1 Problem Definition and Notation

In the unsupervised domain adaptation, one of the domains (source) is supervised $\{\hat{\mathbf{x}}_i, \hat{y}_i\}_{i \in [N^s]}$ with $N^s$ data points $\hat{\mathbf{x}}_i$ and the corresponding labels $\hat{y}_i$ from a discrete set $\hat{y}_i \in \mathcal{Y} = \{1, \dots, Y\}$. The other domain (target), on the other hand is unsupervised and has $N^u$ data points $\{\mathbf{x}_i\}_{i \in [N^u]}$.

We further assume that two domains have different distributions $\hat{\mathbf{x}}_i \sim p_s$ and $\mathbf{x}_i \sim p_t$ defined on the same space $\hat{\mathbf{x}}_i, \mathbf{x}_i \in \mathcal{X}$. We consider a case in which there are two feature functions $\Phi_s, \Phi_t : \mathcal{X} \to \mathcal{R}^d$ applicable to source and target separately. These feature functions extract the information both shared among domains and explicit to the individual ones. The way we model common features is by sharing a subset of parameters between feature functions as $\Phi_s = \Phi_{\theta_c, \theta_s}$ and $\Phi_t = \Phi_{\theta_c, \theta_t}$. We use deep neural networks to implement these functions. In our implementation, $\theta_c$ corresponds to the parameters in the first few layers of the networks and $\theta_s, \theta_t$ correspond to the respective final layers. In general, our model is applicable to any hierarchical and differentiable feature function which can be expressed as a composite function $\Phi_s = f_{\theta_s}(g_{\theta_c}(\cdot))$ for both source and target.

### 3.2 Consistent Structured Transduction

Our method is based on jointly learning the transferable domain specific representations for source and target as well as estimating the labels of the unsupervised data-points. We denote these two main components of our method as transduction and adaptation. The transduction is the sub-problem of labelling unsupervised data points and the adaptation is the sub-problem of solving for the domain shift. In order to solve this joint problem tractably, we exploit two heuristics: cyclic consistency for adaptation and structured consistency for transduction.

**Cyclic consistency:** One desired property of $\Phi_s$ and $\Phi_t$ is consistency. If we estimate the labels of the unsupervised data points and then use these points with their estimated labels to estimate the labels of supervised data-points, we want the predicted labels of the supervised data-points to be consistent with the ground truth labels. Using the inner product as an asymmetric similarity metric as $s(\hat{\mathbf{x}}_i, \mathbf{x}_j) = \Phi_s(\hat{\mathbf{x}}_i)^\mathsf{T} \Phi_t(\mathbf{x}_j)$, this consistency can be represented with the following diagram.

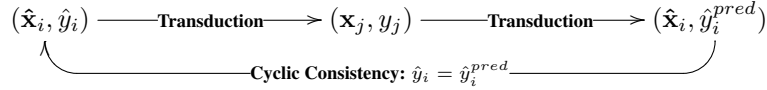

It can be shown that if the transduction from target to source follows a nearest neighbor rule, cyclic consistency can be enforced without explicitly computing $\hat{y}_i^{pred}$ using the large-margin nearest neighbor (LMNN)[35] rule. For each source point, we enforce a margin such that the similarity between the source point and the nearest neighbor from the target with the same label is greater than the similarity between the source point and the nearest neighbor from the target with a different label. Formally; $\Phi_s(\hat{\mathbf{x}}_i)^\mathsf{T} \Phi_t(\mathbf{x}_{i+}) > \Phi_s(\hat{\mathbf{x}}_i)^\mathsf{T} \Phi_t(\mathbf{x}_{i-}) + \alpha$ where $\mathbf{x}_{i+}$ is the nearest target having the same class label as $\hat{\mathbf{x}}_i$ and $\mathbf{x}_{i-}$ is the nearest target having a different class label.

**Structured consistency:** We enforce a structured consistency when we label the target points during the transduction. The structure we enforce is; if two target points are similar to each other, they are more likely to have the same label. To do so, we create a k-NN graph of target points using a similarity metric $\Phi_t(\mathbf{x}_i)^\mathsf{T} \Phi_t(\mathbf{x}_j)$. We denote the neighbors of the point $\hat{\mathbf{x}}_i$ as $\mathcal{N}(\hat{\mathbf{x}}_i)$. We enforce structured consistency by penalizing neighboring points of different labels proportional to their similarity score.

Our model leads to the following optimization problem, over the target labels $y_i$ and the feature function parameters $\theta_c, \theta_s, \theta_t$, jointly solving transduction and adaptation.

$$\min_{\substack{\theta_c, \theta_s, \theta_t, \\ y_1, \dots y_{N^u}}} \underbrace{\sum_{i \in [N^s]} [\Phi_s(\hat{\mathbf{x}}_i)^\mathsf{T} \Phi_t(\mathbf{x}_{i-}) - \Phi_s(\hat{\mathbf{x}}_i)^\mathsf{T} \Phi_t(\mathbf{x}_{i+}) + \alpha]_+}_{\text{Cyclic Consistency}} + \lambda \underbrace{\sum_{i \in [N^u]} \sum_{\mathbf{x}_j \in \mathcal{N}(\mathbf{x}_i)} \Phi_t(\mathbf{x}_i)^\mathsf{T} \Phi_t(\mathbf{x}_j) \mathbb{1}(y_i \neq y_j)}_{\text{Structured Consistency}}$$

$$s.t. \quad i^+ = \arg\max_{j|y_j = \hat{y}_i} \Phi_s(\hat{\mathbf{x}}_i)^\mathsf{T} \Phi_t(\mathbf{x}_j) \quad \text{and} \quad i^- = \arg\max_{j|y_j \neq \hat{y}_i} \Phi_s(\hat{\mathbf{x}}_i)^\mathsf{T} \Phi_t(\mathbf{x}_j)$$

(1)

where $\mathbb{1}(a)$ is an indicator function which is 1 if $a$ is true and 0 otherwise. $[a]_+$ is a rectifier function which is equal to $\max(0, a)$.

We solve this optimization problem via alternating minimization through iterating over solving for unsupervised labels $y_i$(transduction) and learning the similarity metric $\theta_c, \theta_s, \theta_t$ (adaptation). We explain these two steps in detail in the following sections.

### 3.3 Transduction: Labeling Target Domain

In order to label the unsupervised points, we base our model on the k-nearest-neighbor rule. We simply compute the k-NN supervised data point for each unsupervised data point using the learned metric and transfer the corresponding majority label. Formally, given a similarity metric $\theta_c, \theta_s, \theta_t$, the k-NN rule is $(y_i)^{pred} = \arg\max_y \frac{k_y(\mathbf{x}_i)}{k}$ where $k_y(\mathbf{x}_i)$ is the number of samples having label $y$ in the $k$ nearest neighbors of $\mathbf{x}_i$ from the source domain. One major issue with this approach is the inaccuracy of transduction during the initial stage of the algorithm. Since the learned metric will not be accurate, we expect to see some noisy k-NN sets. Hence, we propose two solutions to solve this problem.

**Structured Consistency:** Similar to existing graph transduction algorithms [4, 36], we create a k-nearest neighbor (k-NN) graph over the unsupervised data points and penalize disagreements of labels between neighbors.

**Reject option:** In the initial stage of the algorithm, we let the transduction step use the reject $R$ as an additional label (besides the class labels) to label the unsupervised target points. In other words, our transduction algorithm can decide to not label (reject) some of the points so that they will not be used for adaptation. As the learned metric gets more accurate in the future iterations, transduction algorithm can change the label from R to other class labels.

Using aforementioned heuristics, we define our transduction sub-problem as[1]:

$$\min_{y_1, \ldots y_{N^u} \in \mathcal{Y} \cup R} \sum_{i \in [N^u]} l(\mathbf{x}_i, y_i) + \lambda \sum_{i \in [N^u]} \sum_{\mathbf{x}_j \in \mathcal{N}(\mathbf{x}_i)} \Phi_t(\mathbf{x}_i)^\intercal \Phi_t(\mathbf{x}_j) \mathbb{1}(y_i \neq y_j) \tag{2}$$

where $l(\mathbf{x}_i, y) = \begin{cases} 1 - \frac{k_y(\mathbf{x}_i)}{k} & y \in \mathcal{Y} \\ \gamma \max_{y' \in \mathcal{Y}} \frac{k'_y(\mathbf{x}_i)}{k} & y = R \end{cases}$ and $\gamma$ is relative cost of the reject option.

The $l(\mathbf{x}_i, R)$ is smaller if none of the class has a majority, promoting the reject option for undecided cases. We also modulate the $\gamma$ during learning to decrease number of reject options in the later stage of the adaptation. This problem can approximately be solved using many existing methods. We use the $\alpha$-$\beta$ swapping algorithm from [5] since it is experimentally shown to be efficient and accurate.

### 3.4 Adaptation: Learning the Metric

Given the predicted labels $y_i$ for unsupervised data points $\mathbf{x}_i$, we can then learn a metric in order to minimize the loss function defined in (1). Following the cyclic consistency construction, the LMNN rule can be represented using the triplet loss defined between the supervised source data points and their nearest positive and negative neighbors among the unsupervised target points. We do not include the target-data points with reject labels during this construction. Formally, we can define the adaptation problem given unsupervised labels as;

$$\min_{\theta_c, \theta_s, \theta_t} \sum_{i \in [N^s]} [\Phi_s(\hat{\mathbf{x}}_i)^\intercal \Phi_t(\mathbf{x}_{i-}) - \Phi_s(\hat{\mathbf{x}}_i)^\intercal \Phi_t(\mathbf{x}_{i+}) + \alpha]_+ + \lambda \sum_{i \in [N^u]} \sum_{\mathbf{x}_j \in \mathcal{N}(\mathbf{x}_i)} \Phi_t(\mathbf{x}_i)^\intercal \Phi_t(\mathbf{x}_j) \mathbb{1}(y_i \neq y_j)$$
$$\tag{3}$$

where

$$i^+ = \arg\max_{j|y_j = \hat{y}_i} \Phi_s(\hat{\mathbf{x}}_i)^\intercal \Phi_t(\mathbf{x}_j) \quad \text{and} \quad i^- = \arg\max_{j|y_j \neq \hat{y}_i, y_j \neq R} \Phi_s(\hat{\mathbf{x}}_i)^\intercal \Phi_t(\mathbf{x}_j) \tag{4}$$

We optimize this function via stochastic gradient descent using the sub-gradients $\frac{\partial loss}{\partial \theta_s}, \frac{\partial loss}{\partial \theta_t}$ and $\frac{\partial loss}{\partial \theta_c}$. These sub-gradients can be efficiently computed with back-propagation *(see [1] for details)*.

## 3.5 Implementation Details

We use Alexnet [17] and LeNet [18] architectures with small modifications. We remove their final softmax layer and change the size of the final fully connected layer according to the desired feature dimension. We consider the last fully connected layer as domain specific $(\theta_s, \theta_t)$ and the rest as common network $\theta_c$. Common network weights are tied between domains, and the final layers are learned separately. In order to have a fair comparison, we use the same architectures from [11] only modifying the embedding size. *(See supplementary material [1] for details).*

Since the office dataset is quite small, we do not learn the network from scratch for office experiments and instead we initialize with the weights pre-trained on ImageNet. In all of our experiments, we set the feature dimension as 128. We use stochastic gradient descent to learn the feature function with AdaGrad[8]. We initialize convolutional weights with truncated normals having std-dev 0.1, biases with constant value 0.1, and use a learning rate of $2.5 \times 10^{-4}$ with batch size 512. We start the rejection penalty with $\gamma = 0.1$ and linearly increase with each epoch as $\gamma = \frac{\#epoch-1}{M} + 0.1$. In our experiments, we use $M = 20$, $\lambda = 0.001$ and $\alpha = 1$.

---

**Algorithm 1** Transduction with Domain Shift

**Input:** Source $\hat{\mathbf{x}}_{1 \cdots N^s}, \hat{y}_{1, \cdots N^s}$, Target $\mathbf{x}_{1, \cdots, N^u}$,
Batch Size $2 \times B$
**for** $t = 0$ **to** max_iter **do**
    Sample $\{\hat{\mathbf{x}}_{1,...,B}, \hat{y}_{1,...,B}\}, \{\mathbf{x}_{1,...,B}\}$
    Solve (2) for $\{y_{1...B}\}$
    **for** $i = 1$ **to** $B$ **do**
        **if** $\hat{y}_i \in y_{1...B}$ **and** $\exists k \; y_k \in \mathcal{Y} \setminus \hat{y}_i$ **then**
            Compute $(i^+, i^-)$ using $\{y_{1...B}\}$ in (4)
            Update $\frac{\partial loss}{\partial \theta_{\mathbf{c}}}, \frac{\partial loss}{\partial \theta_{\mathbf{s}}}, \frac{\partial loss}{\partial \theta_{\mathbf{t}}}$
        **end if**
    **end for**
    $\eta(t) \leftarrow$ Adagrad Rule [8]
    $\theta_{\mathbf{c}} \leftarrow \theta_{\mathbf{c}} + \eta(t)\frac{\partial loss}{\partial \theta_{\mathbf{c}}}, \theta_{\mathbf{s}} \leftarrow \theta_{\mathbf{s}} + \eta(t)\frac{\partial loss}{\partial \theta_{\mathbf{s}}},$
    $\theta_{\mathbf{t}} \leftarrow \theta_{\mathbf{t}} + \eta(t)\frac{\partial loss}{\partial \theta_{\mathbf{t}}}$
**end for**

---

# 4 Experimental Results

We evaluate our algorithm on various unsupervised domain adaptation tasks while focusing on two different problems: hand-written digit classification and object recognition.

**Datasets:** We use MNIST [19], Street View House Number [21] and the artificially generated version of MNIST -MNIST-M- [11] to experiment our algorithm on the digit classification task. MNIST-M is simply a blend of the digit images of the original MNIST dataset and the color images of BSDS500 [2] following the method explained in [11]. Since the dataset is not distributed directly by the authors, we generated the dataset using the same procedure and further confirmed that the performance is the same as the one reported in [11]. Street View House Numbers is a collection of house numbers collected from Google street view images. Each of these three domains are quite different from each other. Among many important differences, the most significant ones are MNIST being grayscale whilw the others are colored, and SVHN images having extra confusing digits around the centered digit of interest. Moreover, all domains are large-scale, having at least 60k examples over 10 classes.

In addition, we use the Office [25] dataset to evaluate our algorithm on the object recognition task. The office dataset includes images of the objects taken from Amazon, captured with a webcam and captured with a D-SLR. Differences between domains include the white background of Amazon images versus realistic webcam images, and the resolution differences. The Office dataset has fewer images, with a maximum of 2478 per domain over 31 classes.

**Baselines:** We compare our method against a variety of methods with and without feature learning. **SA\***[9] is the dominant state-of-the-art approach not employing any feature learning, and **Backprop(BP)**[11] is the dominant state-of-the-art employing feature learning. We use the available source code of [11] and [9] and following the evaluation procedure in [11], we choose the hyper-parameter of [9] as the highest performing one among various alternatives. We also compare our method with the **source only** baseline which is a convolutional neural network trained only using the source data. This classifier is clearly different from our nearest neighbor classifier; however, we experimentally validated that the CNN always outperformed the nearest neighbor based classifier. Hence, we report the highest performing source only method.

**Evaluation:** We evaluate all algorithms in a *fully transductive* setup [12]. We feed training images and labels of first domain as the source and training images of the second domain as the target. We evaluate the accuracy on the target domain as the ratio of correctly labeled images to all target images.

## 4.1 Results

Following the fully transductive evaluation, we summarize the results in Table 1 and Table 2. Table 1 summarizes the results on the object recognition task using office dataset whereas Table 2 summarizes the digit classification task on MNIST and SVHN.

Table 1: Accuracy of our method and the state-of-the-art algorithms on Office dataset.

| SOURCE<br>TARGET | AMAZON<br>WEBCAM | D-SLR<br>WEBCAM | WEBCAM<br>D-SLR | WEBCAM<br>AMAZON | AMAZON<br>D-SLR | D-SLR<br>AMAZON |
|---|---|---|---|---|---|---|
| GFK [12] | .398 | .791 | .746 | .371 | .379 | .379 |
| SA* [9] | .450 | .648 | .699 | .393 | .388 | .420 |
| DLID [6] | .519 | .782 | .899 | - | - | - |
| DDC [33] | .618 | .950 | .985 | .522 | .644 | .521 |
| DAN [20] | .685 | .960 | .990 | .531 | .670 | .540 |
| BACKPROP [11] | .730 | **.964** | **.992** | .536 | .728 | .544 |
| SOURCE ONLY | .642 | .961 | .978 | .452 | .668 | .476 |
| OUR METHOD (K-NN ONLY) | .727 | .952 | .915 | .575 | .791 | .521 |
| OUR METHOD (NO REJECT) | .804 | .962 | .989 | .625 | .839 | .567 |
| OUR METHOD (FULL) | **.811** | **.964** | **.992** | **.638** | **.841** | **.583** |

Tables 1&2 show results on object recognition and digit classification tasks covering all adaptation scenarios. Our experiments show that our proposed method outperforms all state-of-the-art algorithms. Moreover, the increase in the accuracy is rather significant when there is a large domain difference such as MNIST↔MNIST-M, MNIST↔SVHN, Amazon↔Webcam and Amazon↔D-SLR. Our hypothesis is that the state-of-the-art algorithms

Table 2: Accuracy on the digit classification task.

| SOURCE<br>TARGET | M-M<br>MNIST | MNIST<br>M-M | SVHN<br>MNIST | MNIST<br>SVHN |
|---|---|---|---|---|
| SA* [9] | .523 | .569 | .593 | .211 |
| BP [11] | .732 | .766 | .738 | .289 |
| SOURCE ONLY | .483 | .522 | .549 | .162 |
| OUR METHOD(K-NN ONLY) | .805 | .795 | .713 | .158 |
| OUR METHOD(NO REJECT) | .835 | .855 | .774 | .323 |
| OUR METHOD(FULL) | **.839** | **.867** | **.788** | **.403** |

such as [11] are seeking features invariant to the domains whereas we seek an explicit similarity metric explaining both differences and similarities of domains. In other words, instead of seeking an invariance, we seek an equivariance.

Table 2 further suggests that our algorithm is the only one which can successfully perform adaptation from MNIST to SVHN. Clearly the features which are learned from MNIST cannot generalize to SVHN since the SVHN has concepts like color and occlusion which are not available in MNIST. Hence, our algorithm learns SVHN specific features by enforcing accurate transduction in the adaptation.

Another interesting conclusion is the asymmetric results. For example, adapting webcam to Amazon and adapting Amazon to webcam yield very different accuracies. The similar asymmetry exists in MNIST and SVHN as well. This observation validates the importance of an asymmetric modeling.

To evaluate the importance of joint labelling and reject option, we compare our method with self baselines. Our self-baselines are versions of our algorithm not using the reject option (**no reject**) and the version using neither reject option nor joint labelling (**k-NN only**). Results on both experiments suggest that joint labelling and the reject option are both crucial for successful transduction. Moreover, the reject option is more important when the domain shift is large (e.g. MNIST→SVHN). This is expected since transduction under a large shift is more likely to fail a situation that can be prevented with reject option.

### 4.1.1 Qualitative Analysis

To further study the learned representations and the similarity metric, we performed a series of qualitative analysis in the form of nearest neighbor and tSNE[34] plots.

Figure 1 visualizes example target images from MNIST and their corresponding source images. First of all, our experimental analysis suggests that MNIST and SVHN are the two domains with the largest difference. Hence, we believe MNIST↔SVHN is a very challenging set-up and despite the huge

visual differences, our algorithm results in accurate nearest neighbors. On the other hand, Figure 2 visualizes the example target images from webcam and their corresponding nearest source images from Amazon.

The difference between invariance and equivariance is clearer in the tSNE plots of the Office dataset in Figure 3 and the digit classification task in Figure 4. In Figure 3, we plot the distribution of features before and after adaptation for source and target while color coding class labels. We use the learned embeddings as output of $\Phi_s$ and $\Phi_t$ as an input to tSNE algorithm[34]. As Figure 3 suggests, the source domain is well clustered according to the object classes with and without adaptation. This is expected since the features are specifically fine-tuned to the source domain before the adaptation starts. However, the target domain features have no structure before adaptation. This is also expected since the algorithm did not see any image from the target domain. After the adaptation, target images also get clustered according to the object classes.

In Figure 4, we show the digit images of the source and target after the adaptation. In order to see the effect of common features and domain specific features separately, we compute the low-dimensional embeddings of the output of the shared network (output of the first fully connected layer). We further compute the NN points between the source and target using $\Phi_s$ and $\Phi_t$, and draw an edge between NNs. Clearly, the target is well clustered according to the classes and the source is not very well clustered although it has some structure. Since we learn the entire network for digit classification, our networks learn discriminative features in the target domain as our loss depends directly on classification scores in the target domain. Moreover, discriminative features in the target arises because of the transductive modeling. In comparison, state of the art domain invariance based algorithms only try to be invariant to the domains without explicit modeling of discriminative behavior on the target. Hence, our method explicitly models the relationship between the domains and results in an equivarient model while enforcing discriminative behavior in the target.

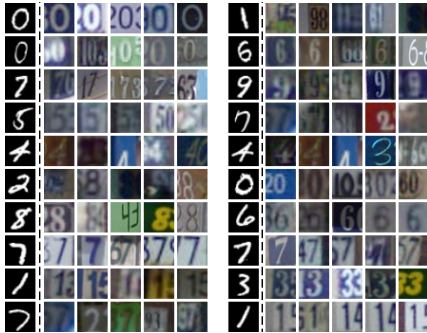

Figure 1: Nearest neighbors for SVHN→MNIST exp. We show an example MNIST image and its 5-NNs.

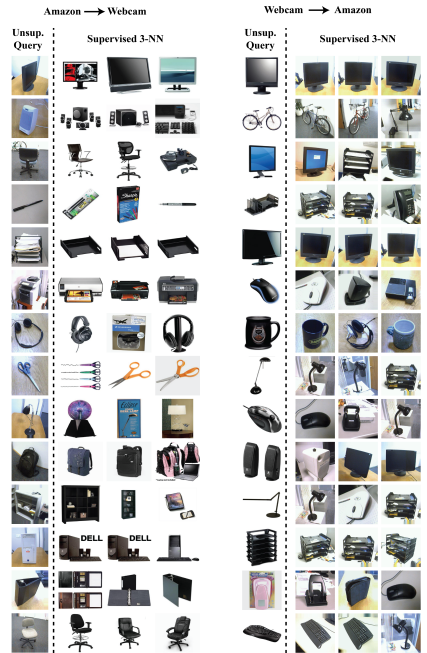

Figure 2: Nearest neighbors for Amazon↔Webcam exp. We show an example Amazon image and its 3-NNs.

## 5 Conclusion

We described an end-to-end deep learning framework for jointly optimizing the optimal deep feature representation, cross domain transformation, and the target label inference for state of the art unsupervised domain adaptation.

Experimental results on digit classification using MNIST[19] and SVHN[21] as well as on object recognition using the Office[25] dataset show state of the art performance with a significant margin.

**Acknowledgments**

We acknowledge the support of ONR-N00014-13-1-0761, MURI - WF911NF-15-1-0479 and Toyota Center grant 1191689-1-UDAWF.

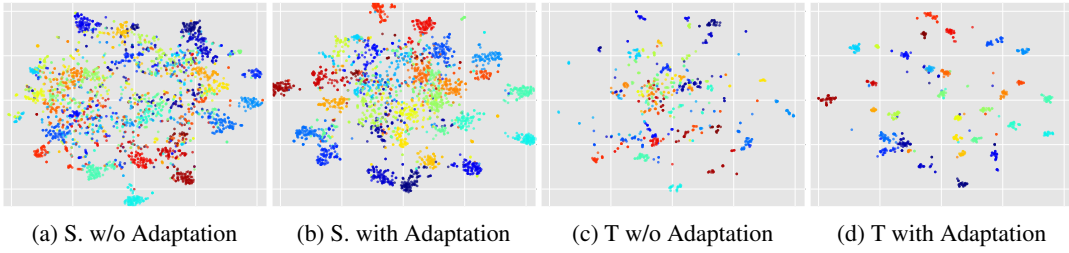

(a) S. w/o Adaptation  (b) S. with Adaptation  (c) T w/o Adaptation  (d) T with Adaptation

Figure 3: tSNE plots for office dataset Webcam(S)→Amazon(T). Source features were discriminative and stayed discriminative as expected. On the other hand, target features became quite discriminative after the adaptation.

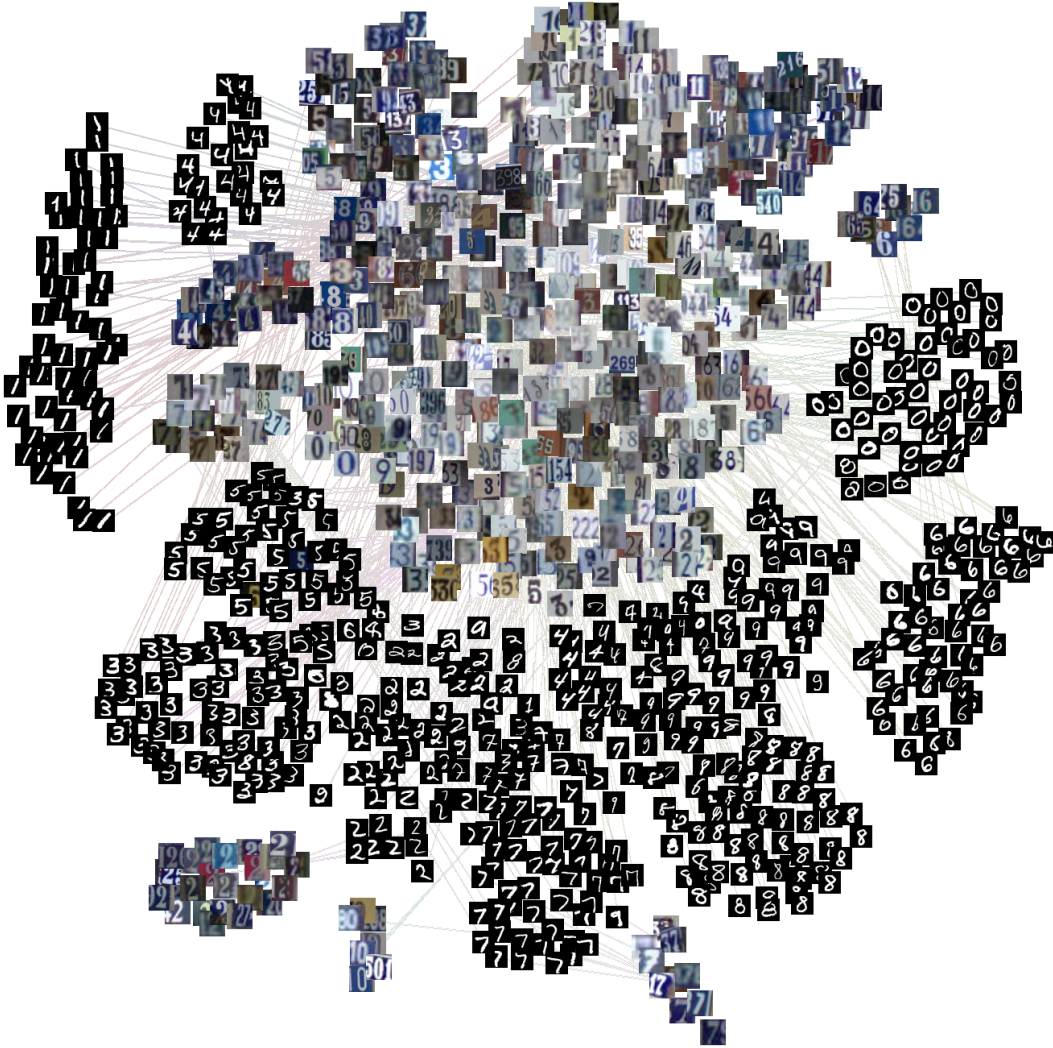

Figure 4: tSNE plot for SVHN→MNIST experiment. Please note that the discriminative behavior only emerges in the unsupervised target instead of the source domain. This explains the motivation behind modeling the problem as transduction. In other words, our algorithm is designed to be accurate and discriminative in the target domain which is the domain we are interested in.

## Footnotes

[1] The subproblem we define here does not directly correspond to optimization of (1) with respect to $y_1, \ldots y_{N^u}$. It is extension of the exact sub-problem by replacing 1-NN rule with k-NN rule and introducing reject option.

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
