[Reviews · NeurIPS 2016]

Reviewer 1

Summary

In this paper, the authors proposed a new approach based on deep learning by incorporating an idea of transductive learning for transfer learning. Experiments are conducted on a fully transductive setup. Though the experimental experimental results look promising, there are some major concerns on the proposed model as well as the experimental setup. Please refer to my detailed comments.

Qualitative Assessment

Regarding the proposed model, below are the major concerns: 1. In Abstract and Introduction, the authors highlighted several times that transfer learning or domain adaptation aims to align the mismatch between the training and testing data distributions, such that good generalization can be obtained across domains or tasks. In the problem setup, the authors further explicitly state that \hat{x}_i and x_i follow different distributions p_s and p_t, respectively. However, different from some existing methods, like [19] and [Pan etal., Domain adaptation via transfer component analysis, IEEE TNN, 2011], the proposed model indeed does not explicitly minimize the distance or align the mismatch between the training and testing distributions. There is no guarantee that based on the new representation, the mismatch issue between distributions can be addressed. Therefore, I would suggest the authors the rephrase those claims carefully. The new high-level representation learned by deep learning may be able to reduce the difference between domains, but is not able to align distributions explicitly. 2. Compared to other transfer learning methods, the new idea here is to introduce a tranductive step to make use of predicted labels on the target domain unlabeled data. This looks new to transfer learning. However, this idea is indeed borrowed from the co-training technique in semi-supervised learning (transductive setting). The adaptation step, which learns common and task-specific parameters, is quite standard in deep learning based transfer learning methods. Thus, technically, I would consider the proposed model is a combination of existing techniques. 3. The solution to labeling target domain unlabeled data, i.e., (2), is quite heuristic. The basic idea of the Reject Opinion is to introduce a kind-of confidence on the predicted labels. However, it would be more solid if some theoretical analysis is provided. 4. Discussions on convergence of the alternating optimization procedure are missing. Regarding experiments, 1. I guess the performance of the proposed method is quite sensitive to the value of k in k-nn on different datasets. Sensitivity analysis on k is missing. In addition, in practice, how to tune the value of k is lack of discussions. 2. I guess the performance of the proposed method is also sensitive to the value of \gamma in the proposed Rejection Option. Sensitivity analysis on k is missing. Similarly, in practice, how to tune the value of \gamma is lack of discussions. 3. Experiments are only conducted on a fully transductive setup, which is not practical. Actually, after the labels of the target domain unlabled data are estimated by the proposed model, a k-nn can be applied to make predictions on out-of-sample target domain test data. It would be more interesting to show results on an out-of-sample test dataset.

Confidence in this Review

2-Confident (read it all; understood it all reasonably well)


Reviewer 2

Summary

This paper propose to solve the domain adaptation problem under the setting that the label of data in the target domain are unknown. To solve this problem, the proposed solution alternative do the transduction and adaptation. Experimental results demonstrate the effectiveness of the proposed solution.

Qualitative Assessment

I have several concerns about the work. Firstly, for the Structured Consistency term, it only enforces the instances with different labels have small similarity, it may result that instances with the same label also have small similarity, which is not desirable for the classification on the target domain. How can the proposed solution avoid this? Secondly, the setting of hyper-parameters. the lambda in equation 1 should greatly affects the performance of the proposed solution. Please shows the performance of the propose solution with respect to different lambda. Please also show the effect of alpha. Thirdly, since the proposed solution do the transduction and adaptation alternatively. Usually how many iterations are needed? It seems for each iteration, we need to learn the DNNs, if there are N iterations, then we need to learn the DNN N times. Therefore the whole framework should be very time consuming. Please report the training time of different methods.

Confidence in this Review

2-Confident (read it all; understood it all reasonably well)


Reviewer 3

Summary

The paper presents an end-to-end deep framework to learn the transferable feature representation for predicting the labels of target domain data-points. The key idea in their proposed method is the iterative optimization strategy, which runs two main components for adaptation and transduction in turn. The authors evaluate their algorithm on several unsupervised domain adaptation tasks and show the good performance on classification and recognition.

Qualitative Assessment

The paper is well written and easy to follow. The authors contribute two interesting heuristics in their optimization strategy: cyclic consistency for domain shift and structured consistency for prediction. Based on my understanding, the former heuristic focuses on the alignment between source and target manifolds in the feature embedding space, and the latter optimizes to group target examples with the same label together. The optimization strategy looks reasonable and the experimental results outperform all state-of-the-art methods. As the authors state in the paper, their approach has one major issue of the inaccuracy of transduction during the initial stage of the algorithm. Two solutions have been proposed to solve this problem, but I still have the following questions for authors: 1) The structured consistency only considers to pull examples with different class labels apart from each other, but it does not optimize to push target data-points from the same classes to each other. Although the cyclic consistency optimizes to penalize two neighboring points of same or different labels, it just enforces in the similarity metric between source and target domain. Why not optimize the similarity metric of two neighboring points of same labels in same domain? 2) I cannot find how the parameters of $\theta_s$ and $\theta_t$ are initialized. Are they initialized randomly? Because the $\theta_s$ and $\theta_t$ are tightly coupled, it will lead to a noisy and even bad start. The authors should provide more details to display how such defective start can convergence to the optimal result, e.g. the accurate curve of the learned metric during the iterations. 3) I think it would be helpful if the authors can provide a convergence curve of the optimization loss. The number of the "max_iter" in Algorithm 1 is also welcome to offer. About spelling: 1) Line 153: the numerator $k'_y(x_i)$ should be $k_{y'}(x_i)$

Confidence in this Review

2-Confident (read it all; understood it all reasonably well)


Reviewer 4

Summary

This paper presents an unsupervised domain-adaptation method that jointly optimize both labels of the target data and representations of the source and target data. The proposed method is based on two types of consistency: cyclic consistency and structured consistency. The cyclic consistency is defined as a consistency between the label of the source data that is predicted based on the inferred label of the target data and its ground-truth. The structured consistency is defined as a consistency between the inferred labels of two target points that are similar to each other. Experimental results show that the proposed method performs better than some state-of-the-art methods.

Qualitative Assessment

The basic idea of this paper is joint optimization of the target label and representation of each domain data. I appreciate that the idea is reasonable, and it should be effective for unsupervised domain adaptation. However, I have some major concerns about its implementation as shown below. In Eq. (2), the first term of Eq. (1) is ignored, however, it would be unreasonable because i+ or i- in the first term of Eq. (1) depends on (y_1, ..., y_Nu). In Eq. (2), the loss l(x,y) is newly introduced. Since it changes the original objective function and doesn't appear in Eq. (3), convergence of the alternating minimization is not already guaranteed. In Eq. (3), the second term of Eq. (1) is ignored, however, it would be unreasonable because Phi_t in the second term of Eq. (1) depends on both theta_t and theta_c. After rebuttal: In the rebuttal, the authors show high stability of the proposed method, and I appreciate that the authors purposefully omit some terms of the object function in the sub-problems to tackle the inaccuracy problem in the initial iterations. However, I feel it somewhat ad-hoc, and I wonder if the optimization process actually minimizes the original object function shown in Eq. 1. I think the authors should modify the object function instead of changing sub-problems. Moreover, the authors describe "It cross-domain distances are accurate ..." in the rebuttal, but it may contradict the above inaccuracy problem.

Confidence in this Review

3-Expert (read the paper in detail, know the area, quite certain of my opinion)


Reviewer 5

Summary

This paper proposed a new deep model for unsupervised domain adaptation. The proposed method replaces softmax layer and loss function with specifically designed domain adaptive objective. The objective jointly models transduction and adaptation. By optimizing the objective, the proposed method achieves state of the art on digit and office benchmark dataset.

Qualitative Assessment

A new deep learning approach for unsupervised domain adaptation is proposed in this paper. The method achieves state-of-the-art performance on digit and office benchmark dataset. The paper is clear and well organized. Though the contribution looks incremental, the experimental performance is promising. 1. Does the alternating solver converge? How stable is the algorithm, and how stable are the results? Could you get the same result every time? 2. line 166, I cannot understand why (3) is convex when feature functions are convex. The feature function is in the minus part of (3) 3. The related work seems insufficient to me. The most recent related work mentioned is [10], which is from ICML 2015. [5][19][32] are compared in experiments without explanation. I think they also need to be discussed in related work section. 4. The writing could be improved, especially in introduction and related work. Many sentences are broken by reference in the related work section.

Confidence in this Review

2-Confident (read it all; understood it all reasonably well)